# Cancer awareness among adolescents in Irish schools: A cross-sectional study

**Stephanie M. Lawrence** ⓘ*, **Mohamad M. Saab, Serena FitzGerald, Josephine Hegarty**

Catherine McAuley School of Nursing and Midwifery, University College Cork, Cork, Ireland

* 116224704@umail.ucc.ie

## Abstract

### Purpose

The aim of this study was to assess adolescents' awareness of cancer signs and symptoms, cancer risk factors, cancer screening programmes, and perceived barriers to seeking medical advice.

### Methods

A cross-sectional survey was conducted using an adapted version of the adolescent cancer awareness tool which was originally modified from the Cancer Awareness Measure (CAM) (Version 2.1). The sample included 474 adolescents aged 15 to 18 years recruited from nine Irish schools between November 2021 and May 2022.

### Results

Awareness of cancer warning signs and symptoms was low when open-ended (recall) questions were used and relatively high for closed (recognition) questions. Unexplained lump or swelling was the most frequently identified cancer symptom. The least reported were difficulty swallowing and a sore that does not heal. Smoking was the most reported cancer risk factor. The least reported were not eating enough fruit and vegetables, a diet high in fat, and infection with viruses. Generally, females had greater awareness than males. The greatest barrier to seeking help was "worry about what the doctor might find" and the least reported barrier was "I don't feel respected by the doctor."

### Conclusions

Overall recognition of symptoms or risk factors of cancer was higher than recall. Several modifiable barriers to medical help-seeking were identified. Findings from this study suggest further exploration using a qualitative approach to investigate the factors influencing adolescents' cancer awareness and barriers to help-seeking.

**Data availability statement:** All relevant data are within the paper and its Supporting Information files.

**Funding:** The authors would like to thank the Marie Keating Foundation for funding aspects of this study. Funding sources played no role in the study's design or conduct, data collection, analysis, interpretation of results, preparation, review, approval of the manuscript, or decision to submit the manuscript for publication.

**Competing interests:** The authors have declared that no competing interests exist.

# 1. Introduction

Adolescence is the life stage between infancy and adulthood, spanning the ages of 10 to 19 years [1]. It is a unique stage of human development characterised by a period of experimentation during which exploratory behaviours can develop into health-risk behaviours. Evidence suggests that exposure to health risk behaviours without adequate protection contributes to lifestyle-related cancer diagnoses [2–4].

Globally, 300,000 new cancer cases are diagnosed among the adolescent population each year [5]. For example, in the United Kingdom (UK), cancer in young adults aged 15 to 24 years accounts for around 2,000 (0.6%) of all cancer cases [6]. In the United States of America, about 5,000 to 6,000 adolescents aged 15 to 19 years are diagnosed with cancer annually [7,8]. According to the National Cancer Registry Ireland, 2,960 children and adolescents aged between 10 and 19 years in both sexes were diagnosed with an all-invasive cancer (minus Nonmelanoma skin cancer) in Ireland between 1994 and 2021 [9]. Leukaemia is the most common cancer in adolescents in Ireland, followed by cancers of the brain and nervous system, sarcoma, and germ cell tumours (e.g., ovarian and testicular cancers.

The term "cancer awareness" has been widely used to explore knowledge of cancer symptoms, risk factors, self-examination, and screening programmes [10–12]. Most studies on cancer awareness involve the adult population as the risks of cancer increase with age; however, it is well known that behavioural factors related to cancer may be acquired during adolescence. In recent years, there has been an increased focus on interventions aimed to promote cancer awareness and health risk behaviours pertaining to cancer among adolescents [4,13,14].

The most suitable period to begin health education is during adolescence, when a person can cognitively comprehend the negative effects of engaging in health risk behaviours [15,16]. Early adoption of healthy lifestyle choices is essential for lowering the lifetime risk of developing cancer [17–19]. Investigating adolescents' awareness of cancer signs and symptoms, cancer risk factors and help seeking behaviour may be of particular importance in developing school-based cancer awareness programmes aimed at reducing the burden of cancer later in life. Therefore, the aim of this study was to assess adolescents' awareness of cancer signs and symptoms, cancer risk factors, cancer screening programmes, and perceived barriers to seeking medical advice. The influence of gender and whether individuals knew someone with cancer on study variables was also explored.

# 2. Methods

## 2.1 Study design

A descriptive cross-sectional study design was used. This study is reported using the Strengthening the Reporting of Observational Studies in Epidemiology (STROBE) checklist for cross-sectional studies [20]. The checklist can be found in S1 File.

## 2.2 Participants

Participants were recruited using a non-probability convenience sampling strategy. The inclusion criteria were adolescents, aged 15 to 18 years, and attending transition year. In Ireland, transition year is a one-year stand-alone programme that forms the first year of a three-year senior cycle. It acts as a bridge between the junior cycle and senior cycle [21].

Participants were sought from public schools, inclusive of schools located in socio-economically deprived areas in Munster, the largest of Ireland's four provinces by land area, and the third largest by population. Schools located in socio-economically deprived areas are referred to as Delivering Equality of Opportunity in Schools (DEIS) schools. The focus of

DEIS is to address the educational needs of children and young people from disadvantaged communities, from pre-school through second-level education (3 to 18 years) [22].

A priori sample size calculation indicated that a sample of 385 students would be necessary to construct a 95% confidence interval (CI) for a single proportion (e.g., proportion who recognises a particular cancer symptom) using the 2016 census indicating a population of 371,588 adolescents aged 13–18 years with a margin of error of 5% (total width of CI = 10%) and assuming a population proportion of 50% [23,24].

### 2.3  Data collection

A list of schools located in Munster was obtained from the Department of Education's website [25]. An email with information about the study was sent to the principal in each school, requesting their expression of interest in the study. Nine schools expressed their interest in taking part in the study. The researcher explained the purpose of the study by telephone then posted an information pack to the school containing a letter about the study, assent forms for adolescents, and consent forms for parents. Transition year course coordinators distributed the forms to students who were interested in taking part. A reminder email was sent to the course coordinators to remind students to have the signed consent and assent forms available on the day of data collection.

Data were collected from November 2021 to May 2022. Data collection took place in-person, in a private room in each of the nine participating schools. The researcher provided a detailed explanation of the study and ensured that all participants provided assent as well as parental consent. Prior to survey completion, all participants were asked not to discuss their answers with other students. All participants remained in the room until everyone had completed the survey. Survey completion took approximately 25 minutes.

### 2.4  Instruments

**2.4.1 Socio-demographic survey.**  The following socio-demographic questions were collected: age, gender, ethnicity, language (English spoken at home and what other languages spoken at home) and whether the adolescent knew a relative or friend who had been diagnosed with cancer.

**2.4.2 Cancer awareness measurement tool.**  We utilised an adapted version of the Cancer Awareness Measure (CAM) [26,27]. Kyle et al. amended the survey to use among a cohort of British adolescents [28,29]. Permission was sought to use the adapted CAM tool a priori for the purpose of our study.

The survey consisted of six distinct sections. The first of which was the socio-demographic section followed by an open-text question asking participants to list as many cancer warning signs and symptoms as they could remember (termed recall). The total number of answers was reported reflecting adolescents' knowledge of cancer signs and symptoms.

Recognition of signs and symptoms of cancer was assessed using nine items. The total number of "yes" answers was then added up to determine the overall score for identifying cancer signs and symptoms.

Perceived barriers to presenting cancer warning symptoms to a doctor were measured using a list of thirteen barriers which were further categorised into emotional, practical, and service level. Responses ranged from "yes often," "yes sometimes," to "no" and "don't know" on a four-point Likert scale and an open-text section to include "other," which were re-categorised for analysis as "yes" or "no." A total score was determined by summing the "yes" responses.

The awareness of known cancer risk factors among adolescents was evaluated using a list of eight cancer risk factors. On a five-point Likert scale, responses ranged from "strongly

disagree" to "strongly agree." Responses of "agree" and "strongly agree" were added to produce an overall risk factor score (out of 11).

The last section explored adolescents' knowledge about cancer screening programmes in Ireland. All responses were reported by gender, and by knowing someone with cancer. Data collection instruments are presented in Table 1.

### 2.5  Ethical considerations

Ethical approval was obtained from the Social Research Ethics Committee in University College Cork (Log 208-2019). All participants were required to sign an assent form and obtain consent from a parent/guardian prior to data collection. Free services available to support adolescents who may be going through a cancer diagnosis or who have a family member going through cancer were enclosed within the study information sheet. Data were stored in accordance with university policies and procedures, and General Data Protection Regulation.

### 2.6  Statistical analysis

Prior to data entry and before the questionnaire was distributed, a codebook was developed for coding the possible responses to the questionnaire items. The coding for CAM was per the CAM authors' coding sheet [27]. All data were entered into IBM SPSS Version 27.0 statistical software (Armonk, NY, USA) and statistical analysis was performed. Categorical variables were described using number and percentage and continuous variables were described using mean and standard deviation (SD). When comparing between two groups (i.e., gender [male/female] and knowing someone with cancer [yes/no]), the independent samples t-test was used for continuous variables and Fisher's exact test was used for categorical variables. The mean difference (MD) and its 95% CI was obtained from the independent samples t-test. All statistical tests were two-sided and a p-value $< 0.05$ was considered statistically significant. The cutoff for excluding surveys due to missing data was $\geq 20\%$ [30].

## 3. Results

### 3.1  Participant characteristics

A total of 477 surveys were collected. Of those, three were excluded due to missing data. Therefore, 474 adolescents participated in this study across nine public schools. The percentage of males and females completing the survey was similar (47.6%, n = 224 and 51.4%, n = 242, respectively). Participants' age ranged from 15 to 18 years with a mean (SD) of 15.7 (0.5) years. Most participants were 16 years old (69.9%, n = 328), white Irish (82.8%, n = 389) and spoke English at home (93.6%, n = 436). Most participants (92.5%, n = 421) reported

**Table 1.  Data collection instruments.**

| Instrument | Source | No of items | Answer options |
|---|---|---|---|
| Sociodemographic questionnaire | Researcher- designed | 6 | Closed questions |
| Knowledge of cancer warning signs and symptoms | CAM | NA | Open questions |
| Knowledge of cancer warning signs and symptoms | CAM | 9 | Dichotomous |
| What would put you off going to the doctor if you had a cancer symptom? | CAM | 13 | Closed questions |
| Things that can increase a person's chance of developing cancer | CAM | 8 | 5-point Likert scale |
| Is there cancer screening in Ireland? | Researcher designed | 3 | Dichotomous |

knowing someone with cancer, including a close family member (69%, n = 314). Socio-demographic characteristics are presented in Table 2.

### 3.2 Recall of cancer signs and symptoms

When completing the survey, adolescents were asked to recall any cancer signs and symptoms they could remember without prompts (termed recall) (Table 3). Some of the most frequently recalled cancer signs and symptoms were unexplained lump or swelling (71.1%, n = 337), followed by pain (27.6%, n = 131) and tiredness/fatigue (23.6%, n = 112). The least reported signs and symptoms (i.e., < 1%) were difficulty swallowing (n = 1) and sore that does not heal (n = 1).

The mean number of signs and symptoms recalled was significantly higher for females than males (mean (SD): 2.4 (1.6) vs 1.8 (1.4) respectively, MD (95% CI): 1.3 (0.9 to 1.6), p < 0.001).

Moreover, the mean number of signs and symptoms recalled was significantly higher for those who knew someone with cancer than those who did not know someone with cancer (mean (SD): 2.2 (1.5) vs 1.2 (1.1) respectively, MD (95% CI): 1.0(0.5 to 1.6), p < 0.001) (Table 3).

Table 2. Sociodemographic characteristics of study participants.

| Characteristic | % | (n) |
|---|---|---|
| **Age in years (n = 469)** | | |
| 15 | 28.6 | (134) |
| 16 | 69.9 | (328) |
| 17 | 1.3 | (6) |
| 18 | 0.2 | (1) |
| Mean (SD) | 15.7 | (0.5) |
| **Gender (n = 471)** | | |
| Male | 47.6 | (224) |
| Female | 51.4 | (242) |
| Other | 1.1 | (5) |
| **Ethnicity (n = 470)** | | |
| White Irish | 82.8 | (389) |
| Other ethnic background[a] | 17.2 | (81) |
| **English spoken at home (n = 466)** | | |
| Yes | 93.6 | (436) |
| No | 6.4 | (30) |
| **Number of languages spoken at home (n = 466)** | | |
| One language | 88.6 | (413) |
| Two languages | 10.3 | (48) |
| Three or more languages | 1.1 | (5) |
| **Know someone with cancer (n = 455)** | | |
| Yes | 92.5 | (421) |
| No | 7.5 | (34) |
| Personal history of cancer | 0.2 | (1) |
| Close family member with cancer | 69.0 | (314) |
| Close friend with cancer | 11.0 | (50) |
| Other | 26.2 | (119) |

[a]n = 6 White Irish traveller; n = 52 Any other white background; n = 3 African; n = 6 Chinese; n = 7 Any other Asian background; n = 7 Other.

**Table 3. Recall of cancer warning signs and symptoms.**

| Warning signs and symptoms (Recall) | Overall (n = 474) | | Gender (n = 466)[a] | | | | | Knowing someone with cancer (n = 455)[b] | | | | |
|---|---|---|---|---|---|---|---|---|---|---|---|---|
| | | | Male (n = 224) | | Female (n = 242) | | | Yes (n = 421) | | No (n = 34) | | |
| | % | (n) | % | (n) | % | (n) | p-value[c] | % | (n) | % | (n) | p-value[c] |
| Lump/Swelling | 71.1 | (337) | 62.9 | (141) | 78.9 | (191) | **<0.001** | 74.1 | (312) | 50.0 | (17) | **0.005** |
| Pain | 27.6 | (131) | 23.2 | (52) | 32.2 | (78) | **0.038** | 28.3 | (119) | 14.7 | (5) | 0.109 |
| Tiredness/Fatigue | 23.6 | (112) | 19.6 | (44) | 28.1 | (68) | **0.039** | 24.7 | (104) | 17.6 | (6) | 0.412 |
| Nausea/Sickness | 15.2 | (72) | 11.2 | (25) | 18.6 | (45) | **0.027** | 16.6 | (70) | 2.9 | (1) | **0.045** |
| Cough/Hoarseness | 13.5 | (64) | 17.0 | (38) | 9.1 | (22) | **0.013** | 13.5 | (57) | 14.7 | (5) | 0.797 |
| Change in appearance of a mole | 13.5 | (64) | 7.1 | (16) | 19.8 | (48) | **<0.001** | 15.2 | (64) | 0.0 | (0) | **0.008** |
| Weight loss | 12.4 | (59) | 7.6 | (17) | 17.4 | (42) | **0.002** | 13.5 | (57) | 2.9 | (1) | 0.104 |
| Bleeding | 9.9 | (47) | 11.6 | (26) | 7.4 | (18) | 0.153 | 10.5 | (44) | 2.9 | (1) | 0.233 |
| Generally unwell | 6.5 | (31) | 8.0 | (18) | 5.0 | (12) | 0.191 | 6.9 | (29) | 5.9 | (2) | 1 |
| Feeling weak | 4.9 | (23) | 5.8 | (13) | 4.1 | (10) | 0.522 | 5.0 | (21) | 2.9 | (1) | 1 |
| Loss of appetite | 3.6 | (17) | 2.7 | (6) | 4.5 | (11) | 0.330 | 3.8 | (16) | 0.0 | (0) | 0.622 |
| Bruising | 3.2 | (15) | 0.4 | (1) | 5.8 | (14) | **0.001** | 3.6 | (15) | 0.0 | (0) | 0.617 |
| Change in bowel/bladder habits | 2.7 | (13) | 2.2 | (5) | 2.5 | (6) | 1 | 2.9 | (12) | 2.9 | (1) | 1 |
| Blurred vision | 0.8 | (4) | 0.0 | (0) | 1.7 | (4) | 0.125 | 1.0 | (4) | 0.0 | (0) | 1 |
| Difficulty swallowing | 0.2 | (1) | 0.0 | (0) | 0.4 | (1) | 1 | 0.2 | (1) | 0.0 | (0) | 1 |
| Sore that does not heal | 0.2 | (1) | 0.0 | (0) | 0.4 | (1) | 1 | 0.2 | (1) | 0.0 | (0) | 1 |
| | mean | (SD) | mean | (SD) | mean | (SD) | p-value[d] | mean | (SD) | mean | (SD) | p-value[d] |
| Number of signs and symptoms recalled | 2.1 | (1.5) | 1.8 | (1.4) | 2.4 | (1.6) | **<0.001** | 2.2 | (1.5) | 1.2 | (1.1) | **<0.001** |

[a]data not available for gender (n=3) and gender Other(n=5).

[b]data on knowing someone with cancer not available for n = 19. "Knowing someone with cancer" is defined as the participant having had cancer themselves or knowing a close family member, a friend or someone else that had cancer.

[c]from Fisher's exact test with participant listed sign or symptom: Yes versus No.

[d]from independent samples t-test.

### 3.3 Recognition of cancer warning signs and symptoms

Adolescents were asked to identify common cancer signs and symptoms from a list that they were presented with (termed recognition) (Table 4). Unexplained lump or swelling (94.1%, n = 446), change in the appearance of a mole (78.3%, n = 371) and pain (78.1%, n = 370) were the most recognised cancer warning signs and symptoms. The least recognised potential sign of cancer was a sore that does not heal (42.6%, n = 202). The mean (SD) number of signs and symptoms (out of 9) recognised was 5.9 (2.1).

The mean number of signs and symptoms recognised was significantly higher for females than males (mean (SD): 6.2(2.0) vs 5.6(2.2) respectively, MD (95% CI): 0.6(0.2 to 1.0), p = 0.001).

However, the mean number of signs and symptoms recognised was not significantly different for those who knew someone with cancer compared to those who did not know someone with cancer (mean (SD): 6.0 (2.1) vs 5.4 (2.0) respectively, MD (95% CI): 0.6 (-0.1 to 1.3), p = 0.108) (Table 4).

### 3.4 Recall of cancer risk factors

Adolescents listed any cancer risk factors they could remember without prompts (recall) (Table 5). Smoking was the most recalled risk factor (78.3%, n = 371) followed by drinking alcohol (65.8%, n = 312). The least reported risk factors (i.e., < 1%) were not eating enough fruit and vegetables (n = 1), a high-fat diet (n = 1) and infection with viruses

**Table 4. Recognition of cancer warning signs and symptoms.**

| | Overall (n = 474) | | Gender (n = 466)[a] | | | | | Knowing someone with cancer (n = 455)[b] | | | | |
|---|---|---|---|---|---|---|---|---|---|---|---|---|
| | | | Male (n = 224) | | Female (n = 242) | | | Yes (n = 421) | | No (n = 34) | | |
| **Warning signs and symptoms (Recognition)** | % | (n) | % | (n) | % | (n) | p-value[c] | % | (n) | % | (n) | p-value[c] |
| Lump/swelling | 94.1 | (446) | 92.4 | (207) | 95.5 | (231) | 0.178 | 95.2 | (401) | 88.2 | (30) | 0.094 |
| Change in appearance of a mole | 78.3 | (371) | 68.3 | (153) | 87.2 | (211) | **<0.001** | 81.0 | (341) | 58.8 | (20) | **0.004** |
| Pain | 78.1 | (370) | 74.1 | (166) | 82.2 | (199) | **0.043** | 78.4 | (330) | 79.4 | (27) | 1 |
| Weight Loss | 71.7 | (340) | 72.3 | (162) | 71.1 | (172) | 0.837 | 72.2 | (304) | 58.8 | (20) | 0.115 |
| Change in bladder/bowel habits | 65.6 | (311) | 62.5 | (140) | 69.0 | (167) | 0.144 | 67.0 | (282) | 52.9 | (18) | 0.131 |
| Bleeding | 65.4 | (310) | 63.4 | (142) | 67.8 | (164) | 0.330 | 65.3 | (275) | 73.5 | (25) | 0.452 |
| Cough/hoarseness | 50.6 | (240) | 50.0 | (112) | 50.8 | (123) | 0.926 | 51.3 | (216) | 50.0 | (17) | 1 |
| Difficulty swallowing | 46.0 | (218) | 39.7 | (89) | 51.7 | (125) | **0.012** | 47.7 | (201) | 38.2 | (13) | 0.372 |
| Sore that does not heal | 42.6 | (202) | 38.4 | (86) | 47.1 | (114) | 0.062 | 43.0 | (181) | 41.2 | (14) | 0.859 |
| | **mean** | **(SD)** | **mean** | **(SD)** | **mean** | **(SD)** | **p-value[d]** | **mean** | **(SD)** | **mean** | **(SD)** | **p-value[d]** |
| Number of signs and symptoms recognised (out of 9) | 5.9 | (2.1) | 5.6 | (2.2) | 6.2 | (2.0) | **0.001** | 6.0 | (2.1) | 5.4 | (2.0) | 0.108 |

[a]data not available for gender (n=3) and gender=Other (n=5).

[b]data on knowing someone with cancer not available for n = 19. "Knowing someone with cancer" is defined as the participant having had cancer themselves or knowing a close family member, a friend or someone else that had cancer.

[c]from Fisher's exact test with Yes vs No/Don't know/Did not answer.

[d]from independent samples t-test.

(unspecified/other) (n = 1). The mean (SD) number of cancer risk factors recalled was 2.4 (1.5).

The mean number of risk factors identified was significantly higher for females than males (mean (SD): 2.6(1.5) vs 2.1(1.4) respectively, MD (95% CI): 0.5(0.3 to 0.8), p < 0.001). In contrast, there was not a statistically significant difference in the mean number of risk factors recalled for those who knew someone with cancer comparted to those who did not know someone with cancer (mean (SD): 2.4 (1.5) vs 2.3 (1.5) respectively, MD (95% CI): 0.1 (−0.4 to 0.6), p = 0.652) (Table 5).

### 3.5 Recognition of cancer risk factors

Smoking any cigarettes was the most recognised cancer risk factor (93.7%, n = 444). The least recognised cancer risk factor was eating less than five portions of fruit and vegetables a day (14.8%, n = 70). The mean (SD) number of cancer risk factors recognised was 3.8 (1.8).

There was not a statistically significant difference between males and females in the mean number of risk factors (out of 8) recognised (mean (SD): 3.8 (1.8) vs 3.7 (1.7) respectively, MD (95% CI): 0.04 (−0.3 to 0.4), p = 0.785). Similarly, there was not a statistically significant difference in the mean number of risk factors (out of 8) recognised for those who knew someone with cancer compared to those who did not know someone with cancer (mean (SD): 3.8 (1.8) vs 3.9 (1.7) respectively, MD (95% CI): −0.2 (−0.8 to 0.4), p = 0.555) (Table 6).

### 3.6 Knowledge of cancer screening programmes in Ireland

Ireland possesses three national screening programmes [31]. Most participants (70.3%, n = 333) were aware of the national breast screening programme. This decreased to 59.1% (n = 280) for the national cervical screening programme, and 35% (n = 166) for the national bowel screening programme.

Females were significantly more aware than males of the national breast and cervical screening programmes. Awareness of the bowel screening programme did not differ significantly by gender (33.9% for females' vs 35.3% for males, p = 0.771). As for knowing/not knowing someone with cancer, there were no significant differences in awareness of breast, cervical, or bowel screening programmes in Ireland between the two groups (p > 0.05 for all) (Table 7).

## 3.7 Barriers to help-seeking

Adolescents were asked to choose from a list of barriers that might prevent them from seeking help from a doctor if they had a symptom of concern. These barriers were categorised into emotional, practical, and service level.

The greatest emotional barrier to seeking help was "worry about what the doctor might find" (68.1%, n = 323), followed by "too scared" (56.8%, n = 269) and "too embarrassed" (55.1%, n = 261). The least reported emotional barrier was "not confident to talk about symptoms" (46.6%, n = 221).

Being "too busy" (32.9%, n = 156) was the most frequently reported practical barrier and "difficult to arrange transport" (23%, n = 109) was the least reported practical barrier. Almost one-third of adolescents (32.7%, n = 155) identified "I don't feel at ease with the doctor" as a

**Table 5. Recall of cancer risk factors.**

| | Overall (n = 474) | | Gender (n = 466)[a] | | | | | Knowing someone with cancer (n = 455)[b] | | | | |
|---|---|---|---|---|---|---|---|---|---|---|---|---|
| | | | Male (n = 224) | | Female (n = 242) | | | Yes (n = 421) | | No (n = 34) | | |
| **Cancer risk factors (Recall)** | % | (n) | % | (n) | % | (n) | p-value[c] | % | (n) | % | (n) | p-value[c] |
| Smoking | 78.3 | (371) | 71.4 | (160) | 83.9 | (203) | **0.002** | 78.9 | (332) | 76.5 | (26) | 0.827 |
| Drinking alcohol | 65.8 | (312) | 56.3 | (126) | 74.8 | (181) | **<0.001** | 66.0 | (278) | 64.7 | (22) | 0.853 |
| Diet (unspecified) | 27.0 | (128) | 24.1 | (54) | 28.9 | (70) | 0.250 | 27.6 | (116) | 26.5 | (9) | 1 |
| Getting sunburnt/exposure to the sun | 23.6 | (112) | 18.3 | (41) | 28.1 | (68) | **0.016** | 24.2 | (102) | 20.6 | (7) | 0.835 |
| Not doing enough exercise/physical activity | 17.5 | (83) | 15.6 | (35) | 18.6 | (45) | 0.461 | 17.6 | (74) | 20.6 | (7) | 0.643 |
| Genes/genetics | 7.6 | (36) | 5.8 | (13) | 9.5 | (23) | 0.165 | 7.8 | (33) | 5.9 | (2) | 1 |
| Family history/having a close relative with cancer/Hereditary | 5.5 | (26) | 6.3 | (14) | 5.0 | (12) | 0.553 | 5.9 | (25) | 2.9 | (1) | 0.710 |
| Being overweight | 4.2 | (20) | 1.8 | (4) | 6.2 | (15) | **0.018** | 4.3 | (18) | 2.9 | (1) | 1 |
| Radiation | 4.0 | (19) | 5.4 | (12) | 2.9 | (7) | 0.241 | 3.8 | (16) | 5.9 | (2) | 0.636 |
| Exposure to another person's cigarette smoke (passive smoking) | 1.7 | (8) | 1.8 | (4) | 1.7 | (4) | 1 | 1.7 | (7) | 2.9 | (1) | 0.465 |
| Pollution | 0.8 | (4) | 1.3 | (3) | 0.4 | (1) | 0.355 | 1.0 | (4) | 0.0 | (0) | 1 |
| Stress | 0.8 | (4) | 0.0 | (0) | 1.7 | (4) | 0.125 | 1.0 | (4) | 0.0 | (0) | 1 |
| Eating red or processed meat | 0.4 | (2) | 0.4 | (1) | 0.4 | (1) | 1 | 0.5 | (2) | 0.0 | (0) | 1 |
| Being underweight | 0.4 | (2) | 0.0 | (0) | 0.8 | (2) | 0.500 | 0.5 | (2) | 0.0 | (0) | 1 |
| Not eating enough fruit and vegetables | 0.2 | (1) | 0.4 | (1) | 0.0 | (0) | 0.481 | 0.2 | (1) | 0.0 | (0) | 1 |
| A high fat diet | 0.2 | (1) | 0.0 | (0) | 0.4 | (1) | 1 | 0.2 | (1) | 0.0 | (0) | 1 |
| Infection with viruses (Unspecified/Other) | 0.2 | (1) | 0.0 | (0) | 0.4 | (1) | 1 | 0.2 | (1) | 0.0 | (0) | 1 |
| | **mean** | **(SD)** | **mean** | **(SD)** | **mean** | **(SD)** | **p-value[d]** | **mean** | **(SD)** | **mean** | **(SD)** | **p-value[d]** |
| Number of risk factors recalled | 2.4 | (1.5) | 2.1 | (1.4) | 2.6 | (1.5) | **<0.001** | 2.4 | (1.5) | 2.3 | (1.5) | 0.652 |

[a]data on gender not available for n = 3 and gender=Other for n = 5. Hence, n = 8 excluded from analysis.

[b]data on knowing someone with cancer not available for n = 19. "Knowing someone with cancer" is defined as the participant having had cancer themselves or knowing a close family member, a friend or someone else that had cancer.

[c]from Fisher's exact test with participant listed cancer risk: Yes vs No.

[d]from independent samples t-test.

barrier. Over half of adolescents (52.7%, n = 250) reported that they would contact their doctor within three days if they had a sign or symptom of cancer.

Over a third of adolescents noted "difficulty to make an appointment" (38.4%, n = 182) as a service-level barrier to help-seeking. The least reported service-level barrier was "difficulty talking to the doctor" (28.9%, n = 137).

Only one practical barrier "other things to worry about" was significantly different between males and females, with females more likely than males to consider it a barrier. For two of the service-level barriers, females were more likely than males to consider them a barrier: "worry about wasting the doctor's time" and "difficulty talking to the doctor" (34.7% vs 23.2% respectively, p = 0.008). Females were also more likely than males to not feel at ease with the doctor and to consider this as a barrier to seeking help. None of the barriers to seeking help were associated with knowing someone with cancer (p > 0.05 for all) (Table 8). However,

**Table 6. Recognition of cancer risk factors.**

| Cancer risk factors (Recognition) | Overall (n = 474) % | (n) | Gender (n = 466)[a] Male (n = 224) % | (n) | Female (n = 242) % | (n) | p-value[c] | Knowing someone with cancer (n = 455)[b] Yes (n = 421) % | (n) | No (n = 34) % | (n) | p-value[c] |
|---|---|---|---|---|---|---|---|---|---|---|---|---|
| Smoking any cigarettes | 93.7 | (444) | 94.6 | (212) | 92.6 | (224) | 0.451 | 93.8 | (395) | 88.2 | (30) | 0.267 |
| Drinking more than 1 unit of alcohol a day | 63.1 | (299) | 60.3 | (135) | 66.1 | (160) | 0.211 | 62.5 | (263) | 73.5 | (25) | 0.267 |
| Being overweight (BMI > 25) | 56.5 | (268) | 60.7 | (136) | 52.5 | (127) | 0.076 | 55.6 | (234) | 67.6 | (23) | 0.209 |
| Getting sunburnt more than once as a child | 52.3 | (248) | 46.0 | (103) | 56.6 | (137) | **0.026** | 53.4 | (225) | 41.2 | (14) | 0.211 |
| Infection with HPV | 41.4 | (196) | 37.5 | (84) | 44.6 | (108) | 0.132 | 41.8 | (176) | 41.2 | (14) | 1 |
| Doing less than 60 mins of moderate physical activity 5 times a week | 32.7 | (155) | 39.7 | (89) | 25.6 | (62) | **0.001** | 32.3 | (136) | 41.2 | (14) | 0.343 |
| Eating red or processed meat once a day or more | 21.9 | (104) | 24.1 | (54) | 19.4 | (47) | 0.260 | 21.1 | (89) | 23.5 | (8) | 0.827 |
| Eating less than 5 portions of fruit and vegetables a day | 14.8 | (70) | 13.8 | (31) | 14.9 | (36) | 0.792 | 14.7 | (62) | 17.6 | (6) | 0.619 |
| | **mean** | **(SD)** | **mean** | **(SD)** | **mean** | **(SD)** | **p-value[d]** | **mean** | **(SD)** | **mean** | **(SD)** | **p-value[d]** |
| Number of risk factors recognised (out of 8) | 3.8 | (1.8) | 3.8 | (1.8) | 3.7 | (1.7) | 0.785 | 3.8 | (1.8) | 3.9 | (1.7) | 0.555 |

[a]data on gender not available for n = 3 and gender=Other for n = 5. Hence, n = 8 excluded from analysis.

[b]data on knowing someone with cancer not available for n = 19. "Knowing someone with cancer" is defined as the participant having had cancer themselves or knowing a close family member, a friend or someone else that had cancer.

[c]from Fisher's exact test with Strongly agree/Agree vs Not sure/Disagree/Strongly disagree/Did not answer.

[d]from independent samples t-test

**Table 7. Awareness of cancer screening programmes in Ireland.**

| Cancer screening programmes in Ireland | Overall (n = 474) % | (n) | Gender (n = 466)[a] Male (n = 224) % | (n) | Female (n = 242) % | (n) | p-value[c] | Knowing someone with cancer (n = 455)[b] Yes (n = 421) % | (n) | No (n = 34) % | (n) | p-value[c] |
|---|---|---|---|---|---|---|---|---|---|---|---|---|
| Breast screening | 70.3 | (333) | 62.5 | (140) | 76.9 | (186) | **0.001** | 71.5 | (301) | 64.7 | (22) | 0.433 |
| Cervical screening | 59.1 | (280) | 50.0 | (112) | 67.8 | (164) | **<0.001** | 59.9 | (252) | 55.9 | (19) | 0.717 |
| Bowel screening | 35.0 | (166) | 35.3 | (79) | 33.9 | (82) | 0.771 | 34.2 | (144) | 41.2 | (14) | 0.455 |

[a]data on gender not available for n = 3 and gender=Other for n = 5. Hence, n = 8 excluded from analysis.

[b]data on knowing someone with cancer not available for n = 19. "Knowing someone with cancer" is defined as the participant having had cancer themselves or knowing a close family member, a friend or someone else that had cancer.

[c]from Fisher's exact test with Yes vs No/Don't know/Did not answer.

**Table 8. Barriers to seeking help.**

| Barriers to seeking help | Overall (n = 474) % | (n) | Gender (n = 466)[a] Male (n = 224) % | (n) | Female (n = 242) % | (n) | p-value[c] | Knowing someone with cancer (n = 455)[b] Yes (n = 421) % | (n) | No (n = 34) % | (n) | p-value[c] |
|---|---|---|---|---|---|---|---|---|---|---|---|---|
| **Emotional barriers** | | | | | | | | | | | | |
| Worried about what the doctor might find | 68.1 | (323) | 56.7 | (127) | 79.8 | (193) | **<0.001** | 69.4 | (292) | 52.9 | (18) | 0.056 |
| Too scared | 56.8 | (269) | 42.9 | (96) | 69.8 | (169) | **<0.001** | 57.2 | (241) | 50.0 | (17) | 0.473 |
| Too embarrassed | 55.1 | (261) | 45.5 | (102) | 64.0 | (155) | **<0.001** | 55.3 | (233) | 52.9 | (18) | 0.858 |
| Not confident to talk about symptoms | 46.6 | (221) | 37.9 | (85) | 55.4 | (134) | **<0.001** | 47.7 | (201) | 38.2 | (13) | 0.372 |
| **Practical barriers** | | | | | | | | | | | | |
| Too busy | 32.9 | (156) | 34.8 | (78) | 31.4 | (76) | 0.490 | 33.3 | (140) | 32.4 | (11) | 1 |
| Other things to worry about | 32.7 | (155) | 28.6 | (64) | 37.2 | (90) | **0.050** | 32.8 | (138) | 23.5 | (8) | 0.341 |
| Difficult to arrange transport | 23.0 | (109) | 19.6 | (44) | 25.6 | (62) | 0.150 | 23.3 | (98) | 17.6 | (6) | 0.530 |
| **Service barriers** | | | | | | | | | | | | |
| Difficult to make an appointment | 38.4 | (182) | 36.2 | (81) | 40.1 | (97) | 0.392 | 39.0 | (164) | 35.3 | (12) | 0.718 |
| Worried about wasting the doctor's time | 37.6 | (178) | 28.1 | (63) | 47.1 | (114) | **<0.001** | 38.7 | (163) | 29.4 | (10) | 0.359 |
| Difficult to talk to doctor | 28.9 | (137) | 23.2 | (52) | 34.7 | (84) | **0.008** | 29.9 | (126) | 14.7 | (5) | 0.075 |
| **Other barriers** | | | | | | | | | | | | |
| I don't feel at ease with the doctor | 32.7 | (155) | 23.7 | (53) | 41.3 | (100) | **<0.001** | 33.5 | (141) | 23.5 | (8) | 0.261 |
| I don't feel respected by the doctor | 8.6 | (41) | 6.3 | (14) | 11.2 | (27) | 0.072 | 9.0 | (38) | 5.9 | (2) | 0.756 |

[a]data on gender not available for n = 3 and gender=Other for n = 5. Hence, n = 8 excluded from analysis.

[b]data on knowing someone with cancer not available for n = 19. "Knowing someone with cancer" is defined as the participant having had cancer themselves or knowing a close family member, a friend or someone else that had cancer.

[c]from Fisher's exact test with Yes often/Yes sometimes vs No/Don't know/Did not answer.

adolescents indicated that they would seek medical help for a symptom they thought might be cancer within 24 hours. See S2 File.

## 4. Discussion

Cancer is a major contributor to premature death and increased disease burden in adolescents and young adults globally [32]. This highlights the importance of measuring and addressing cancer awareness including identifying personal risk factors in the adolescent population. According to Abraham et al. adolescence is a particular time in one's life that provides a unique "window of opportunity for cancer risk education and intervention" pg. 51 [33].

In the current study, cancer awareness was higher in those who knew someone with cancer compared to those who did not know someone with cancer; a finding consistent with other studies [29,34,35]. However, findings from our study indicate that, overall, adolescents had low cancer awareness. For instance, while they were aware of "lump or swelling" as a sign and symptom of cancer; they were generally unaware of "a sore that does not heal." This is consistent with other studies [28,29,34]. There are several reasons why adolescents may not be cancer aware and are therefore less likely to recognize some potential signs of cancer. Adolescents may feel invincible at a young age and may not think that they are at risk [36]. This can potentially lead to a lack of concern about warning signs like skin changes. Sores that do not heal may also be mistaken for common skin changes experienced during puberty such as acne, making it challenging to differentiate between normal and abnormal skin changes [37]. Reasons why adolescents generally fail to recognise some known cancer signs and symptoms need to be explored more thoroughly.

Current study findings indicate that many adolescents were aware of the risk factors of "smoking" and "drinking alcohol" but lacked knowledge in relation to other risk factors like not eating enough fruits and vegetables, a high-fat diet, and infection with viruses. This pattern is also apparent in other studies [19,38]. Smoking and vaping are common substance abuse behaviours among adolescents and smoking cessation advertisements are an effective means of encouraging adolescents to quit and prevent smoking [39].

Physical activity levels reduce the risk of cancer by helping control weight, regulating hormone levels, and boosting the immune system [40]. Similarly, a diet rich in fruits and vegetables is associated with a lower risk of developing cancer [41]. However, findings in this study highlight concerns that adolescents are unaware of the benefits of diet and exercise in cancer prevention. Reasons for this need to be explored more thoroughly, however, according to Niehoff et al. physical inactivity and its link to cancer may not be emphasized in public health education; therefore, adolescents may not receive adequate information about the importance of physical activity in cancer prevention [42].

Adolescent face social and financial challenges, which might result in inequities in access to appropriate health care, timely diagnosis, and treatment [32]. Most adolescents in our study reported that they would seek medical help within the first 24 hours for symptoms of concern. Barriers to seeking medical help were categorised into emotional, practical, and service level. Adolescents in this study suggested "worry", as an emotional barrier to help-seeking. Practical barriers related to having "other things to worry about," and over a third of adolescents reported "difficulty to make an appointment" and "difficulty talking to the doctor" as service-level barriers. Adolescents' reasons for not visiting their family physician/general practitioner (GP) are complex [43]. According to a systematic review by Lawrence et al. adolescents often encounter difficulties with confidentiality, trust, and uncertainty when using GP-led primary care services, which can cause reluctance and poor rapport with GPs [44]. Factors highlighted among adolescents included fears, anxiety [45] and feelings of embarrassment [43,46]. These factors make it challenging for adolescents to seek help from a GP for signs and symptoms of concern. Similarly, in a cross-sectional study by Martinović et al. adolescents demonstrated discomfort in sharing information pertaining to health issues (e.g., sexual health and mental health) with other people and tended to use personal sources, such as electronic sources and seek health information from their mother, sister, and female friends [47].

Participants' awareness of screening programmes in Ireland varied. Females were more aware of the national breast and cervical screening programmes in comparison to males. However, over half of study participants were unaware of the human papillomavirus (HPV) virus and its link with cancer. This finding is a cause for concern, considering the WHO published a report on the World Health Assembly on the Global Strategy for the Elimination of Cervical Cancer, adopting a gender-neutral HPV vaccine approach to reduce HPV infections transmitted among the adolescent population, combatting misinformation, minimizing vaccine-related stigma, and promoting gender equity [48]. Of note, the increased prevalence of HPV infections has been linked to increased levels of sexual activity among adolescents and young people aged 10 to 24 years [49]. Men who have sex with men have sexual debut at an average age of 15 years which increases the risk of contracting the HPV virus and anal dysplasia [50].

Research has found that low awareness of potential cancer signs and symptoms is associated with help-seeking delays [51–53]. Potential causes of this delay include but are not limited to negative beliefs about cancer outcomes [14,35,54] and low awareness of health risk behaviours linked to cancer [14,33,38]. In addition, a person is typically more likely to seek medical attention for cancer symptoms that are more obvious, like

swelling, in comparison to less obvious or vague symptoms such as fatigue and weight loss [34,55]. In general, cancer awareness in this study was higher among females compared to males. Similar findings have been reported in other studies which also used the CAM tool [28,34]. Previous studies have also found that female adolescents tend to use health-related applications more frequently [56] and were more willing to search for sensitive health information online [57], in comparison to their male counterparts. Men's help-seeking behavior for cancer symptoms was found to be associated with their level of knowledge and awareness of cancer [58]. According to a systematic review by Fish et al. early detection of symptoms significantly influences men's inclination to seek assistance [58]. Somatic symptoms capture attention, and demand interpretation, however, little is known about how adolescents and young people make sense of somatic symptoms within uncertain health contexts or how the process guides health behaviours and healthcare-initiating behaviours [59].

Cancer awareness programmes should be designed using several ways rather than through a single method. In addition, health promotion programmes need to focus on young males as they are more likely than females to be involved in risk taking behaviour such as alcohol and drug use, high risk sexual behaviour, and engage in fewer health promoting behaviours than females [47,60]. Evidence suggests that adolescents' help-seeking behaviours could be improved by increasing cancer communication [14]. This would help increase cancer awareness, giving adolescents the ability to understand healthcare information and make appropriate health decisions including seeking help.

## Implications for practice, education, and future research

The adoption of positive lifestyle choices at a young age is fundamental to reducing the risk of developing cancer later in life [14,34]. Adolescence is a period where health risk behaviours are developed. Therefore, adolescents are deemed a suitable target for cancer education [4,61].

Cancer awareness interventions need to incorporate strategies to address adolescents' health and help-seeking concerns. Such interventions need to consider the least recognised cancer signs, symptoms, and risk factors including bleeding, a sore that does not heal, inadequate intake of fruits and vegetables, and inadequate physical activity.

Using a school-based cancer awareness intervention poses an opportunity to raise adolescents' cancer awareness and address barriers to seeking medical help. A collaboration between healthcare authorities and educational authorities would be beneficial, incorporating cancer awareness interventions in the curriculum and providing tailored education to adolescents. Cancer awareness strategies that appeal to adolescents ought to be used. These include but are not limited to cancer exhibitions, mobile applications, and other interactive platforms.

Making a GP a prominent feature of cancer awareness programmes might help improve awareness of cancer signs and symptoms, cancer risk factors, and help-seeking behaviour. Moreover, if combined with school cancer awareness programmes, GPs and adolescent interactions could help alleviate barriers in accessing primary care services and improve cancer awareness levels as part of early cancer prevention.

Adolescents' voices must be heard, and their opinions taken into consideration while developing such programmes. This highlights the potential benefits of an in-depth qualitative study with adolescents exploring how cancer awareness programmes can be improved or promoted; which organisations are most suitable to promote such programmes; which learning strategies can be used to promote cancer awareness; and what role the schools/teachers and GPs play in raising cancer awareness among adolescents.

## Strengths and Limitations

Key strengths from our study include the use of the validated CAM tool to collect data, calculating sample size a priori, and seeking participation from nine schools.

The primary limitation of this study is the use of non-probability convenience sampling, seeking participants from one province in Ireland, and recruiting a relatively homogenous sample. This restricts the ability to draw population-level conclusions. Some schools may have had existing relationships with organisations that deliver cancer awareness interventions. This could have led to higher cancer awareness among some participants in these schools.

A further limitation of the study is the retrospective nature of the assessment on recall of cancer warning signs, symptoms, and risk factors. Although we have tried to control for recall bias, it is impossible to avoid it completely, as some participants may have had an opportunity to go back to the recall questions and add in additional signs and symptoms and risk factors outlined in the recognition section.

## Conclusions

In conclusion, this study showed that adolescent school goers have overall low awareness of cancer warning signs, symptoms, and risk factors and experience several emotional, structural, and service-level barriers to help-seeking. Gender differences in cancer awareness and help-seeking exist. This requires further exploration. Study findings reinforce the importance of school-based initiatives to help adolescents recognise cancer warning signs, symptoms, and risk factors and to seek help accordingly, with an emphasis on signs, symptoms, and risk factors that were least recognised. Such initiatives would help reduce or avoid the emergence of health disparities later in life. Understanding the patterns of cancer awareness and the mechanisms that explain the connections between awareness and help-seeking warrant further longitudinal research. Adolescent-specific messages on cancer should be encouraged to address their needs, while considering factors such as gender, age, socio-economic factors, and close contact with individuals who have a cancer diagnosis. Findings from this study could be explored further using in-depth qualitative interviews with adolescents. Such interviews have the potential to explore how cancer awareness programmes can be improved or promoted with a focus on schools.

## Supporting information

**S1 File. Strengthening the Reporting of Observational Studies in Epidemiology (STROBE) checklist for cross-sectional studies.**
(DOCX)

**S2 File. Length of time before seeking medical help for a symptom you thought might be cancer.**
(DOCX)

## Author contributions

**Conceptualization:** Stephanie M. Lawrence, Mohamad M. Saab, Serena FitzGerald, Josephine Hegarty.

**Investigation:** Stephanie M. Lawrence.

**Methodology:** Stephanie M. Lawrence, Mohamad M. Saab, Serena FitzGerald, Josephine Hegarty.

**Supervision:** Mohamad M. Saab, Serena FitzGerald, Josephine Hegarty.

**Visualization:** Stephanie M. Lawrence.

**Writing – original draft:** Stephanie M. Lawrence.

**Writing – review & editing:** Mohamad M. Saab, Serena FitzGerald, Josephine Hegarty.

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
