## [Decision Letter · Decision Letter 0]

30 Dec 2024

PONE-D-24-39876Cancer awareness among adolescents in Irish schools: A cross-sectional studyPLOS ONE

Dear Dr. Lawrence,

Thank you for submitting your manuscript to PLOS ONE. After careful consideration, we feel that it has merit but does not fully meet PLOS ONE’s publication criteria as it currently stands. Therefore, we invite you to submit a revised version of the manuscript that addresses the points raised during the review process.

**ACADEMIC EDITOR: **

Dear Authors, The manuscript needs minor revisions.

Respond point by point to the requests of the reviewers.

Best regards

We look forward to receiving your revised manuscript.

Kind regards,

Omar Enzo Santangelo

Academic Editor

PLOS ONE

Journal Requirements:

Reviewers' comments:

Reviewer's Responses to Questions

**Comments to the Author**

1. Is the manuscript technically sound, and do the data support the conclusions?

Reviewer #1: Yes

Reviewer #2: Yes

2. Has the statistical analysis been performed appropriately and rigorously? 

Reviewer #1: Yes

Reviewer #2: Yes

3. Have the authors made all data underlying the findings in their manuscript fully available?

Reviewer #1: Yes

Reviewer #2: Yes

4. Is the manuscript presented in an intelligible fashion and written in standard English?

Reviewer #1: Yes

Reviewer #2: Yes

5. Review Comments to the Author

Reviewer #1: This research is important because cancer education for adolescents is necessary for effective cancer prevention, and their interest in cancer is influenced by cultural and social factors. I think it would be even better if some points were supplemented.

The purpose of this study is to assess perceptions of adolescents. From the description in 1. Introduction, it seems that the researchers define “adolescents” as those in their teens and early twenties, but the subjects of this study are those who aged 15 to 18 years. Therefore, how about adding this point to the Limitations of the study?

In the “2.3 Data Collection” section, it is stated that the course coordinator distributed the questionnaire to students who were interested in. It is possible that the subjects who were already interested in cancer. I think it would be better to add this bias to the Limitations, too.

In the Discussion, it is stated that “cancer awareness was higher in those who knew someone with cancer compared to those who did not know someone with cancer”, but it is unclear what they are more aware of cancer. It is stated that the number of recalled cancer signs and symptoms were predominantly higher in those who knew someone with cancer than those who did not, but there was no significant difference in awareness of risk factors or screening programs. If this means that people who know cancer patients are more concerned about symptoms, it should be stated as such. Alternatively, if the small number of subjects who do not know cancer patients (92.5% who know and 7.5% who do not) makes it difficult to find a statistically significant difference, it would be less misleading to add that.

I hope these comments are helpful.

Reviewer #2: Abstract:

Results: It is better to provide the values of the main findings included within the results part of the abstract.

Conclusion: Consider rewriting the conclusion part of the abstract taking into account the principal findings of this study.

Keywords, some keywords need to be removed (cross-sectional) and replaced by more related ones

6. PLOS authors have the option to publish the peer review history of their article (what does this mean? ). If published, this will include your full peer review and any attached files.

**Do you want your identity to be public for this peer review?** For information about this choice, including consent withdrawal, please see our Privacy Policy .

Reviewer #1: No

Reviewer #2: **Yes: ** Dr. Anmar AL-TAIE

---

## [Author Response · Author response to Decision Letter 1]

27 Jan 2025

Cancer awareness among adolescents in Irish schools: A cross-sectional study

We would like to thank the Editor and Reviewers of PLOS ONE for their valuable feedback regarding the manuscript above. Below are our point-by-point responses to all comments. Changes in the text are made using the colour red.

EDITOR

Journal Comment Our Response Page and line number

Thank you for submitting your manuscript to PLOS ONE. After careful consideration, we feel that it has merit but does not fully meet PLOS ONE’s publication criteria as it currently stands. Therefore, we invite you to submit a revised version of the manuscript that addresses the points raised during the review process.

Dear Authors,

The manuscript needs minor revisions.

Respond point by point to the requests of the reviewers.

Kind regards,

Omar Enzo Santangelo

Academic Editor

PLOS ONE Thank you for the opportunity to publish in the PLOS ONE Journal.

REVIEWER -1-

Reviewer Comment Our Response Page and line number

This research is important because cancer education for adolescents is necessary for effective cancer prevention, and their interest in cancer is influenced by cultural and social factors. I think it would be even better if some points were supplemented. Thank you for your feedback and suggestion. Please see the following amendments.

The purpose of this study is to assess perceptions of adolescents. From the description in 1.

Introduction, it seems that the researchers define “adolescents” as those in their teens and early twenties, but the subjects of this study are those who aged 15 to 18 years. Therefore, how about adding this point to the Limitations of the study? Thank you for your suggestion this has now been added to the limitations section

While adolescence spans the ages of 10 to 19 years (1), the majority of current study participants were 15- and 16-year-olds. This limits the generalisability of findings to younger and older adolescents.. P32

In the “2.3 Data Collection” section, it is stated that the course coordinator distributed the questionnaire to students who were interested in. It is possible that the subjects who were already interested in cancer. I think it would be better to add this bias to the Limitations, too. Thank you for your suggestion this has now been added to the limitations section

The use of non-probability convenience sampling, coupled with transition year course coordinators distributing the questionnaire to students who expressed interest in participating, increases the likelihood that only students with an interest in and/or prior knowledge of cancer chose to complete the questionnaire. This could potentially introduce bias. P32

In the Discussion, it is stated that “cancer awareness was higher in those who knew someone with cancer compared to those who did not know someone with cancer”, but it is unclear what they are more aware of cancer. It is stated that the number of recalled cancer signs and symptoms were predominantly higher in those who knew someone with cancer than those who did not, but there was no significant difference in awareness of risk factors or screening programs. If this means that people who know cancer patients are more concerned about symptoms, it should be stated as such. Alternatively, if the small number of subjects who do not know cancer patients (92.5% who know and 7.5% who do not) makes it difficult to find a statistically significant difference, it would be less misleading to add that. Thank you for pointing this our and our apologies for this overgeneralization. We have included the following paragraph for clarification as recommended: In the current study, the small number of participants who did not know someone with cancer (i.e., 92.5% knew someone with cancer and 7.5% did not know someone with cancer) makes it difficult to find a statistically significant difference across outcomes. That said, the mean number of signs and symptoms of cancer recalled was significantly higher for those who knew someone with cancer than those who did not know someone with cancer; a finding consistent with other studies (27,31,32).

REVIEWER -2-

Reviewer Comment Our Response Page and line number

Abstract:

Results: It is better to provide the values of the main findings included within the results part of the abstract.

Conclusion: Consider rewriting the conclusion part of the abstract taking into account the principal findings of this study.

Keywords, some keywords need to be removed (cross-sectional) and replaced by more related ones Thank you for your suggestions. The values of the main findings have now been included within the results part of the abstract.

Thank you for your suggestion. The principal findings of this study have now been addressed in the conclusion section of the abstract which we have re-written as follows:

Adopting healthy lifestyles in adolescence is crucial for reducing cancer risk later in life. School-based cancer awareness interventions should address least recognised signs and risk factors. Collaboration between healthcare and education authorities is key, incorporating tailored strategies that appeal to and are co-developed with adolescents. Involving General Practitioners in these programs could reinforce awareness and improve access to and help-seeking in primary care settings. There is also a need for qualitative research to explore, in depth, factors influencing adolescents’ cancer awareness and barriers to help-seeking.

Thank you for your comment. We revised the keywords while using MeSH terms as follows: Adolescent; awareness; health education; help-seeking behavior; neoplasms; risk factors; schools; surveys and questionnaires; signs and symptoms. P2

P3

---

## [Decision Letter · Decision Letter 1]

30 Jan 2025

Cancer awareness among adolescents in Irish schools: A cross-sectional study

PONE-D-24-39876R1

Dear Dr. Lawrence,

We’re pleased to inform you that your manuscript has been judged scientifically suitable for publication and will be formally accepted for publication once it meets all outstanding technical requirements.

Kind regards,

Omar Enzo Santangelo

Academic Editor

PLOS ONE

Additional Editor Comments (optional):

Reviewers' comments:

Reviewer's Responses to Questions

**Comments to the Author**

1. If the authors have adequately addressed your comments raised in a previous round of review and you feel that this manuscript is now acceptable for publication, you may indicate that here to bypass the “Comments to the Author” section, enter your conflict of interest statement in the “Confidential to Editor” section, and submit your "Accept" recommendation.

Reviewer #1: (No Response)

Reviewer #2: (No Response)

2. Is the manuscript technically sound, and do the data support the conclusions?

Reviewer #1: (No Response)

Reviewer #2: Yes

3. Has the statistical analysis been performed appropriately and rigorously? 

Reviewer #1: (No Response)

Reviewer #2: Yes

4. Have the authors made all data underlying the findings in their manuscript fully available?

Reviewer #1: (No Response)

Reviewer #2: Yes

5. Is the manuscript presented in an intelligible fashion and written in standard English?

Reviewer #1: (No Response)

Reviewer #2: Yes

6. Review Comments to the Author

Reviewer #1: (No Response)

Reviewer #2: (No Response)

7. PLOS authors have the option to publish the peer review history of their article (what does this mean? ). If published, this will include your full peer review and any attached files.

**Do you want your identity to be public for this peer review?** For information about this choice, including consent withdrawal, please see our Privacy Policy .

Reviewer #1: **Yes: ** Akiko Fukawa

Reviewer #2: **Yes: ** Dr. Anmar AL-TAIE

---

## [Editor Report · Acceptance letter]

PONE-D-24-39876R1

PLOS ONE

Dear Dr. Lawrence,

I'm pleased to inform you that your manuscript has been deemed suitable for publication in PLOS ONE. Congratulations! Your manuscript is now being handed over to our production team.

Kind regards,

on behalf of

Dr. Omar Enzo Santangelo

Academic Editor

PLOS ONE